# Significance of Lung Ultrasound in Patients with Suspected COVID-19 Infection at Hospital Admission

**DOI:** 10.3390/diagnostics11060921

**Published:** 2021-05-21

**Authors:** Holger Gutsche, Thomas G. Lesser, Frank Wolfram, Torsten Doenst

**Affiliations:** 1Clinic for Thoracic and Vascular Surgery, SRH Wald-Klinikum Gera, 07548 Gera, Germany; Holger.Gutsche@SRH.de (H.G.); Thomas.Lesser@SRH.de (T.G.L.); 2Department of Cardiac and Thoracic Surgery, University Hospital Jena, 07740 Jena, Germany; Doenst@med.uni-jena.de

**Keywords:** lung ultrasound, COVID-19, POCUS

## Abstract

With a lung ultrasound (LUS) the typical findings are interstitial pneumonia. COVID-19 pneumonia is often manifested in sub-pleural areas, which is preferably detected by sonography. An RT-PCR test cannot always ensure a safe differentiation of COVID-19- and non-diseased cases. Clinically challenging is that a reliable and time efficient decision regarding COVID-19 suspects requiring isolation. Therefore, this study was aimed at evaluating the significance of LUS in symptomatic patients with COVID-19 suspicion at hospital admission. A total of 101 patients admitted to a suspect ward with COVID-19-typical symptoms were assessed. All patients received prospectively a standardized LUS at admission. Patients were classified as LUS-positive and -negative cases based on a specific LUS score. The RT-PCR test in combination with the clinical findings served as a reference. Correctly classified were 14/15 COVID-19 diseased suspects as LUS-positive (sensitivity: 93.3%). Twenty-seven out of 61 non-positive cases were classified as false positive with LUS (specificity: 55.7%). In 34/35 patients who were assessed as LUS negative, no COVID-19 disease was detected during the hospitalization. The PPV and NPV of the LUS were 34.1% and 97.1%. LUS is a valuable tool in symptomatic patients for the assessment of COVID-19-disease. The high negative predictive value of LUS is helpful to rule out the disease.

## 1. Introduction

The COVID-19 pandemic poses a major challenge to the health care system. Among other things, the timely and safe differentiation between symptomatic patients with COVID-19 disease and those not infected by Sars-COV-2, is a challenge particularly for hospitalized patients. Due to the very non-specific symptoms such as fever, cough, cold, or unclear infection parameters, a high number of patients without COVID-19 disease are admitted to isolated COVID-wards. Especially elderly, often multi-morbid, patients are thereby exposed to an additional risk of infection, and treatment of their underlying disease and comorbidities might be delayed. To this end, medical resources are tied up in such cases, and this increases the strain upon the health care system in a pandemic situation.

With currently available testing methods, notably the RT-PCR test, the absence of Sars-Cov2 infection can only identified correctly and expeditiously to a limited extent. The false negative rate of the RT-PCR test is being reported up to 38% [1], which can arise as a result of poor sampling quality, improper collection, or unfavorable transport conditions [2]. Furthermore, due to limited test capacities, results can be delayed for several days, or positive test results might be present only after the second or third sample [3]. Therefore, a single negative RT-PCR test result on respiratory specimens is insufficient to rule out COVID-19 disease [4,5,6].

Symptomatic COVID-19 disease is dominantly associated with pneumonia. For imaging and monitoring of those, chest X-Ray (CXR) is of minor importance compared with computed tomography (CT) because of missing specificity [7]. Lung CT of COVID-19 pneumonia shows typical findings of interstitial pneumonia with a sensitivity of 94% [8]. It is recommended in case of urgent COVID-19 suspicion and a negative RT-PCR. However, the examination is not practicable as an incoming screening examination due to high patient volume in a pandemic situation, transport problems, hygiene guidelines, and associated radiation exposure.

The lung ultrasound is becoming a more and more important point-of-care imaging modality in the differential diagnostics of pulmonary and pleural diseases [9]. In the case of interstitial pneumonia, typical ultrasound findings such as an irregular or fragmented pleural line, B-lines, white lung syndrome (WLS), or consolidations can be collected [10]. Since COVID-19 pneumonia manifests mainly peripheral/subpleural, an ideal prerequisite for the use of LUS is available [11]. Early stages of pneumonia can be detected with ultrasound, and the findings correlate well with those by computed tomography [12,13,14]. Due to its portability, LUS allows bedside imaging in absence of radiation and could be a valuable diagnostic screening tool for the early detection or exclusion of patients with Sars-COV-2 infections. 

Therefore, this study is aimed to investigate the significance of LUS in symptomatic patients with COVID-19 suspicion at the time of hospitalization. For this, the diagnostic value of a lung ultrasound with reference to the RT-PCR test with the clinical findings was determined.

## 2. Materials and Methods

### 2.1. Patients

In the period from April to June 2020, patients with clinical suspicion of COVID-19 disease underwent prospective and standardized LUS after admission to the COVID-19-suspect ward of the SRH Wald-Klinikum Gera. The symptoms for inclusion were cough, dyspnea, unclear fever (>38 °C), as well as an increase in laboratory infection parameters (CrP > 10 mg/L; PCT > 0.5 mg/L). Patient characteristics were separately documented from the digitally stored patient data after anonymization. These included ages, gender, symptoms, computed tomography of the chest (if done), and laboratory findings, as well as secondary diseases.

Inclusion criteria were an age over 18 years, no pregnancy, and LUS examination within 48 h after hospital admission. Exclusion was performed in need of immediate ICU transfer and an uncooperative condition.

### 2.2. Lung Ultrasound Examination

All examinations were carried out by one experienced LUS investigator with a background of six years in thoracic and vascular surgery with daily practice of general and lung sonography. The examiner received prior to the study a general LUS training according to EFSUMB and AIUM recommendations and COVID-19 specific online training.

The systematic LUS examination was performed bedside on six thoracic areas of the right (R) and left (L) hemi-thorax in an insulating room at the COVID-suspect ward. 

The investigation was conducted on the basis of international recommendations for COVID-19 lung ultrasound examination [15,16]. Herein the probe was placed onto the intercostal place to ensure imaging without rib shadowing. The investigator scanned for suspicious or pathological LUS feature within each areal.

The following chest areas were investigated:R/L 1: on mid-clavicular line below the clavicula;R/L 2: on mid-clavicular line next to internipple line;R/L 3: on mid axillary line above the internipple line;R/L 4: on mid axillary line below the internipple line;R/L 5: on paravertebral line below the scapula (sitting);R/L 6: on paravertebral line above the diaphragm (sitting).

The examinations were performed with a clinical sonography scanner (LOGIQ/VENUE, GE HealthCare, Solingen, Germany) using a linear probe (9L) or in obese patients with a convex probe (C1-5). The scanner specific lung ultrasound pre-set was used with a median MI of 1.1 (0.8–1.2) and TIS of 0.1 (0.0–0.2). The operator adjusted the placement of the single focus onto the pleural line. LUS frames of five seconds were initially recorded from each areal to ensure the monitoring of at least one breath cycle. The video sequences were evaluated blindly by two LUS experienced investigators (FW, HG) independent of each other, and without knowledge of patients’ condition including the RT-PCR status. In the case of different assessments, a third clinician (TL) was consulted for consensus formation. The definitive classification of all 12 lung areas was carried out by means of a lung score of 0–3 based on the international COVID-19 LUS scoring system according to Soldati et al. [16]:Score 0: inconspicuous continuous pleural line, possible A-lines, visible or invisible B-lines in a number ≤ 3 per field of view;Score 1: B-lines (number > 3) or white lung syndrome (WLS), irregular pleural line or when pleura appears as thickened on sonography (thickened pleural);Score 2: pleural fragmentation with possible sub-pleural small consolidations (<1 cm);Score 3: larger consolidation >1 cm with or without aero-bronchogram.

Patients were classified as “LUS-COVID-positive” if a score of 2 or 3 was present in one area out of 12. A classification as “LUS-COVID-negative” was made when a score of 0 or 1 was found throughout all areas. The patient was excluded from the study if more than one area was missing or could not be assessed.

### 2.3. RT-PCR Reference

The patients suspected for COVID-19 disease already received a RT-PCR test in the emergency department (ED) at admission. In the case of negative test results and continued symptomatic course, a repeated RT-PCR test was carried out by sampling an irritant sputum or throat rinse water. As soon as a positive single test was present during the course of the disease, the patient was classified as “COVID-19-diseased”. All RT-PCR tests were carried out by SYNLAB Medical Care Center Gera (SYNLAB MVZ, Weiden, Germany). A “first RT-PCR” test was defined if taken and successfully analyzed at the ED in the moment of admission. Out of 76 included cases a first RT-PCR test was successfully performed on 68 cases. Four RT-PCR samples could not be evaluated due to material defects and four patients were admitted to the COVID-19 ward due to a positive ambulatory COVID-19 test without detailed documentation.

### 2.4. Statistical Analysis

The statistical analysis was performed using MedCalc (Vers. 19.1.7 Medcalc LTD, Ostend, Belgium). Quantitative variables are presented as median and Inter Quartile Range (25th percentile, 75th percentile), and categorical as counts and proportion. Statistical evaluation was performed on the categorical variables with the Fisher’s Exact–Boschloo test, and on continuous data with Mann–Whitney U tests. The tests’ power was estimated based on the resulting counts [17]. In addition, sensitivity, specificity, positive predictive value (PPV), negative predictive value (NPV), and diagnostic accuracy were evaluated for the LUS score, first RT-PCR Test, and in a combination of both. The positive count for combination of LUS and RT-PCR test was made when an LUS-COVID-positive or positive first RT-PCR test result was present. The estimates are reported with 95% confidence interval (CI). A two-sided *p*-value < 0.05 was defined as significant.

## 3. Results

### 3.1. Patient Characteristics

A total of 101 patients with clinical suspicion of COVID-19 disease were examined using LUS. After excluding 25 cases, 76 patients were included for the study. A summary of studies patient selection is shown in Figure 1.

Baseline characteristics, secondary diseases and clinical symptoms of the study patients are presented in Table 1. The median age was 75.5 years. Both genders were almost equally distributed (male 48.7% and female 51.3%). The median length of stay on the COVID-19 suspect ward was 4.7 days for all patients, 2.4 days for non-COVID-19-diseased and 13.9 days for COVID-19-diseased patients. In 15 (19.7%) of the patients, COVID-19 disease was finally diagnosed, in 61 patients (80.3%) COVID-19 disease had been excluded after the end of the diagnostics. With regard to the incidence of fever, cough, or dyspnea as well as blood gases pO_2_ and pCO_2_, no differences were found between COVID-19 and non-COVID-19-diseased patients (*p* > 0.05). The loss of taste and/or sense of smell were found significantly more commonly found among patients with confirmed COVID-19 disease (*p* = 0.047). No significant difference with regard to the secondary diseases was present between non- and COVID-19-diseased cases.

### 3.2. Diagnostic Value of Lung Ultrasound and in Combination with RT-PCR Test

The LUS examination was performed at the COVID-suspect ward within a median time delay of 18 h 5 min (9 h 48 min–23 h 30 min) after admission to the hospital’s ED. Fourteen out of the fifteen COVID-19-diseased patients were classified LUS-COVID-positive (sensitivity: 93.3%; CI: 68.1% to 99.8%). Of 61 non-COVID-19-diseased patients, 27 LUS-COVID-positive cases were false-positive classified with LUS (specificity: 55.7%; CI: 42.5% to 68.5%). In 34 patients who were assessed as negative by LUS (LUS-COVID-negative), no COVID-19 disease was detected in the further course of the hospitalization. One patient was classified LUS-COVID-negative in the presence of COVID-19 disease and therefore judged to be false-negative. This patient was admitted with symptomatic cough and an increased infection parameter (CrP = 47 mg/L) due to a fracture from an external, not COVID-19 treating, hospital. No fever or dyspnea and lung infiltrations on CXR were present.

The positive and negative predictive value (PPV, NPV) of the LUS was 34.2% (CI: 27.5% to 41.5%) and 97.1% (CI: 83.5% to 99.6%), respectively. The first RT-PCR test, carried out at hospital admission such as in the ED (68 of 72 RT-PCR tests), showed a sensitivity of 90.9% (CI: 58.7% to 99.8%), a specificity of 100% (CI: 93.7% to 100%), resulting in a PPV of 100% and an NPV of 98.3% (CI: 89.8% to 99.7%). Until a reliable result was available, approximately 2.1 (CI: 1.9 to 2.3) RT-PCR tests were performed, therefore non-COVID-19-diseased patients were discharged from the COVID-19 ward after 2.4 days (CI: 2.0 to 2.9). The result of the first RT-PCR test combined with the LUS-COVID classification, revealed a sensitivity of 100% (CI: 71.5% to 100%) and thus a negative predictive value of 100% (Table 2).

### 3.3. Pathological Lung Ultrasound Findings

More than two B-lines, white lung syndrome, pleural fragmentation, and consolidations were found significantly more frequently in COVID-19-diseased than in non-COVID-19-diseased patients. When COVID-19 was present, more than two B-lines appear in all patients, WLS in 66.7%, subpleural consolidation in 53.3%, and pleural fragmentation in 93.3%. COVID-19 patients were significantly more often classified as LUS-COVID-positive than non-COVID-19-diseased cases (93.3% vs. 44.3%, *p* < 0.01). For COVID-19-diseased patients an LUS score of 3 was found in 53.3%, a score of 2 in 40%, whereas only one patient (1.6%) showed an LUS score 1, characterized by a thickened pleural line in three areas and increased B-lines in two areas. Pleural effusion was found more frequently in non-COVID-19-diseased (23%) than in COVID-19-diseased (6.7%) patients which revealed no statistical significance. A summary of LUS score and pathological findings are presented in Table 3. 

The most common LUS findings in non-COVID-19-diseased patients were thickened pleura (98.4%), increased B-lines (75.4%), and a pleural fragmentation (42.6%). Subpleural consolidation as a typical finding in interstitial pneumonia was shown in 26.2%. LUS scores of 3 and 2 were present in 26.2% and 18.0%, respectively. An LUS score 1 was significantly more common in non-COVID-19-diseased than in COVID-19-diseased patients (54.1% vs. 6.7%).

Non-pathological, aerated lung in all areas (score 0) was not present in all COVID-19-diseased, and in only one non-COVID-19-diseased (1.6%) patient, which was not significant.

## 4. Discussion

We were able to show that the early lung ultrasound examination of inpatient admission provides a diagnostic gain and is valuable in the clarification of Sars-COV-2 suspected patients at hospital admission. With LUS, 93.3% of the COVID-19-positive cases could be classified correctly. In 97.1% of the LUS-negative cases, no COVID-19 disease was present. The NPV increases to 100% when the LUS findings are combined with the first RT-PCR test, carried out at hospital admission.

In contrast, symptomatic screening (fever, cough, dyspnea, blood gas) is insufficient, as our baseline characteristic shows. COVID-19 specific symptoms are slightly more frequently found on diseased patients, which was not significant. Although the loss of taste and smell is a significant symptom it is not applicable for screening due to the low incidence (6.8%). 

The duration on a COVID suspect ward (2.4 days) for non-COVID-19-diseased cases is signally inappropriate, causing a delay of therapy, risk of infection, and an unnecessary consumption of resources. The RT-PCR test shows a valuable diagnostic accuracy for the study specific cohort of symptomatic suspects at hospital admission. However, the LUS provides a higher sensitivity than the first RT-PCR test, but with a much lower specificity. Using LUS in combination with an RT-PCR test could help reducing the isolation time on a suspect ward, and therefore reduce the strain on the health care system in a pandemic situation.

The specificity of the LUS examination is low (55.7%), as the sonographic findings are not only COVID-19 specific. Such pathological LUS signs can occur in other disease-causing interstitial pneumonia, or are present due to underlying chronical lung disease, as frequently found in patients requiring hospitalization. Therefore, LUS cannot provide an etiological diagnosis. Comparable results were found in a recent study by Sorlini et al. [18], reporting a sensitivity and specificity of 92.0% and 64.9%, respectively. The higher specificity can be explained by the higher prevalence of the COVID-19 disease (74.7%) in the cohort studied. 

In addition, Volpicelli et al. [19] reported an LUS sensitivity of 90.2% and specificity of 50.5% investigating early LUS under various inclusion criteria. Narinx et al. [20] found an LUS sensitivity of 93.3% and specificity of 21.3% for patients at ED admission. Highly interesting, is the work of Pivetta et al. [21], with a high LUS sensitivity (94.4%) and specificity (95%). In contrast to our work, all the above studies classified the appearance of increased B Lines and WLS as a COVID-19 related LUS sign, which would represent a score of 1 in our study. This choice, different inclusion criteria, and the clinical situation of the LUS examiner being aware of the patients’ condition (not blinded), might explain the high specificity found by Pivetta et al. However, these studies in addition to our results, demonstrate the potential of LUS, but also the need for a deeper definition of COVID-19 specific LUS features and for an appropriate definition of inclusion criteria.

The LUS score classification has a decisive influence on the test results of the LUS. A lower cut off, that includes score 1 to be LUS-COVID-positive, leads to a high sensitivity and high false positive rate. LUS as an early diagnostic method of non-COVID-19 diseased cases should be focused on the exclusion of the disease to avoid hospital admission. Our chosen cut off provides a justifiable NPV, which is to our knowledge the highest one reported so far. Therefore, we recommend expressing suspicion of COVID-19 disease only with an LUS score of ≥2 (Figure 2) for patients in early hospital admission. In our study, a high prevalence of cardiac disease (82%) and hypertension (80%) were present in the non-COVID-19-disease group, which are often associated with increased lung water. These underlying pathologies frequently manifest pathological LUS artefacts such as B-lines and WLS (score 1) [9].

When assessing LUS findings, the presentation of clinical symptoms as well as the stage of the disease should be considered. There is likely to be a weak correlation between the severity of symptoms and the specificity of LUS signs. As found in studies with non-critical and mild symptomatic COVID-19 patients, the characteristic sonographic manifestations show increased numbers of B-lines as well as sub-pleural consolidation [22]. In our experience with hospitalized symptomatic patients, sub-pleural consolidations and pleural fragmentation are more typical LUS findings in the presence of COVID-19, even if detected in only one lung areal. 

Furthermore, the timing of the ultrasound examination with regard to the disease stage, impacts the availability of the LUS features. If applied at the recovery stage, pathological LUS findings may be diminished. Thereby, the false negative case in our study could be explained. The patient was admitted from the local hospital due to a fracture and cough. The COVID-19 disease might already have been in remission, which was further indicated by the negative CXR that showed an absence of lung infiltration.

During COVID-19 disease, the histopathological features may vary and coexist, dependent upon the time-point in disease evolution, and the severity of disease [23]. Different phases of the alveolar damage were described, such as pre-exudative, exudative, organizing, and fibrotic phases. We believe that the LUS features can correspond approximately to the histopathological phases. Sonographic appearance of thickened pleura and increased B-line counts are signs at the onset of the disease, whilst pleural fragmentation and consolidations occur more frequently in the later stages. 

Herein it was noticed that linear probes visualize pleural irregularities more clearly than curved probes, and therefore represent our first choice.

One unspecific LUS finding, is the pleural effusion which was initially considered as a pathological COVID-19 finding [24]. However, as found in this study, pleural effusion was more frequently present in non-COVID-19-diseased patients and appeared only in one COVID-19-positive case, which confirms the findings of other studies [14,25].

The limitations of our study are primarily the low number of cases. The results should be verified on a larger patient sample size. However, a sufficient power (>0.8) was estimated for exact testing, based on the proportions found in the study. The significance found for pathological B-Lines should be considered with caution, due to instability of Fischer’s exact test when zero counts are present [26].

In order to optimize the triage of incoming symptomatic patients in need for hospitalization, early LUS should be recommended for use in the ED. Despite the relatively long inclusion criteria of 48h after admission, most of the LUS exams were performed within the first day (median delay 18 h). A corresponding prospective study is recommended. The evaluation of the LUS findings and the classification into a score system are subjective and examiner dependent. Using a retrospective assessment by three experienced LUS clinicians, we have tried to minimize this subjective influence. The study included patients with clinical symptoms or infection parameters that led to hospitalization. Whether patients in the outpatient setting with low symptoms also have the described LUS findings, cannot be answered with the present study.

## 5. Conclusions

The manifestation of COVID-19 pneumonia in the peripheral/subpleural lung areas, is an ideal condition for the application of LUS. Point-of-care ultrasound of the lung in patients with suspected COVID-19 disease should play a key role at the initial examination. The high negative predictive value of LUS is helpful for the exclusion of the infection and is further improved in combination with a simultaneous negative RT-PCR test. LUS can be helpful to minimize the risk of COVID-19 infection due to unfounded admission to COVID-19 wards.

## Figures and Tables

**Figure 1 diagnostics-11-00921-f001:**
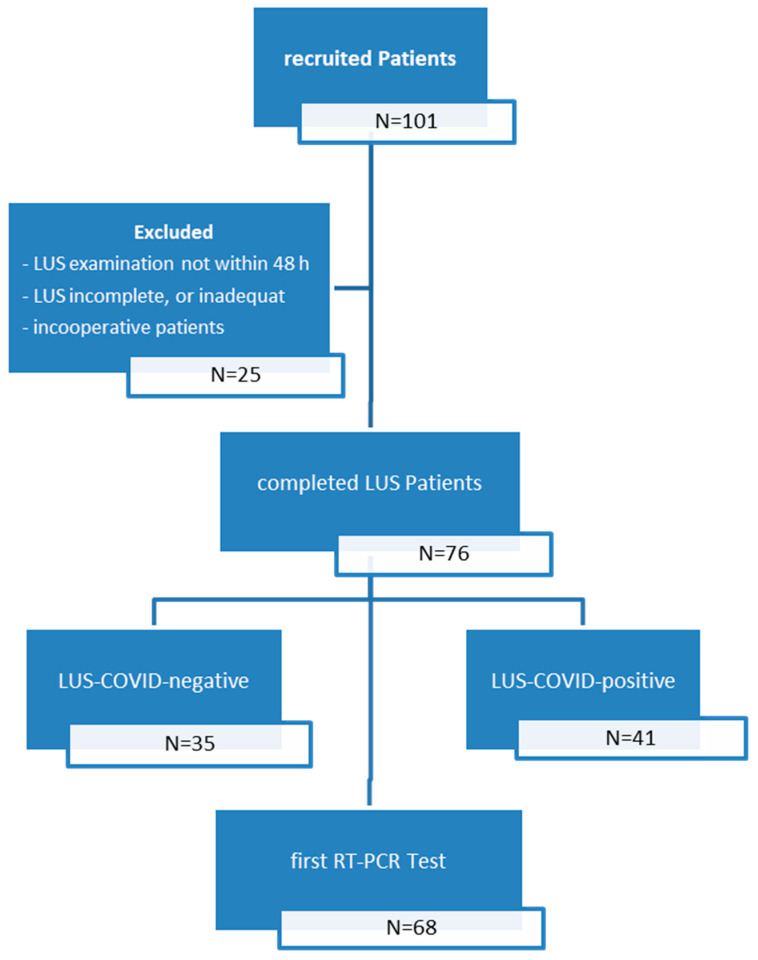
Flow chart of the patient selection and classification according to the lung ultrasound and reference standard.

**Figure 2 diagnostics-11-00921-f002:**
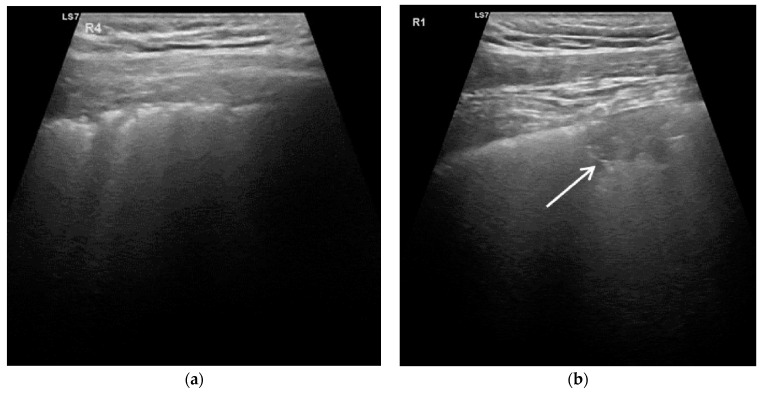
Lung ultrasonography image of a patient with COVID-19 disease: (**a**) pronounced fragmentation of the pleural line and small subpleural consolidations (according to LUS score 2) and (**b**) lung ultrasonography image of a patient with COVID-19 disease. Thickened pleural line with irregularity and a large consolidation with air bronchogram (arrow; according to LUS score 3). Corresponding video frames are available online (Appendix A).

**Table 1 diagnostics-11-00921-t001:** Baseline characteristics of the study patients with secondary diseases and clinical symptoms.

Variables		Patients N = 76	COVID-19-Diseased N = 15 (19.7%)	Non-COVID-19 Diseased N = 61 (80.3%)	*p*
Age, years		75.5	83	74	0.56
		(68 to 81)	(58 to 89)	(68 to 81)	
Sex					0.78
Male		37 (48.7%)	8 (53.3%)	29 (47.5%)	
Female		39 (51.3%)	7 (46.7%)	32 (52.5%)	
Secondary diseases					
Cardiac diseases	Yes	61 (80.3%)	11 (73.3%)	50 (82.0%)	0.48
	No	15 (19.7%)	4 (26.7%)	11 (18.0%)	
Pulmonary diseases	Yes	26 (34.2%)	5 (33.3 %)	21 (34.4%)	1.0
	No	50 (65.8%)	10 (66.7%)	40 (65.6%)	
Diabetes mellitus	Yes	26 (34.2%)	5 (33.3%)	21 (34.4%)	1.0
	No	50 (65.8%)	10 (66.7%)	40 (65.6%)	
Hypertension	Yes	60 (78.9%)	11 (73.3 %)	49 (80.3%)	0.72
	No	16 (21.1%)	4 (26.7%)	12 (19.7%)	
Symptoms at hospital admission					
Fever on intake	Yes	38 (50.0%)	8 (53.3%)	30 (49.2%)	0.78
	No	38 (50.0%)	7 (46.7%)	31 (50.8%)	
Cough	Yes	32 (42.1%)	8 (53.3%)	24 (39.3%)	0.39
	No	44 (57.9%)	7 (46.7%)	37 (60.7%)	
Dyspnea	Yes	39 (51.3%)	8 (53.3%)	31 (50.8%)	1.00
	No	37 (48.7%)	7 (46.7%)	30 (49.2%)	
Loss of taste and/or sense of smell	Yes	5 (6.6%)	3 (20.0%)	2 (3.3%)	<0.05
	No	71 (93.4%)	12 (80.0%)	59 (96.7%)	

A *p*-value < 0.05 was defined as significant.

**Table 2 diagnostics-11-00921-t002:** Diagnostic significance of LUS score classification, first RT-PCR test, and in combination of both in COVID-19 suspected patients.

	Sensitivity (%)	Specificity (%)	PPV (%)	NPV (%)	Accuracy (%)
LUS	93.3	55.7	34.2	97.1	63.2
First RT-PCR test	90.9	100	100	98.3	98.5
Combination of LUS and first RT-PCR test	100	56.14	30.6	100	63.2

LUS, lung ultrasound; PPV, positive predictive value; NPV, negative predictive value.

**Table 3 diagnostics-11-00921-t003:** Classification results and lung ultrasound characteristics of study patients.

Test			All Counts % Column	COVID-19-Diseased	Non-COVID-19-Diseased	*p*-Value
LUS COVID-positive			41 (53.9%)	14 (93.3%)	27 (44.3%)	<0.01
LUS COVID-negative			35 (46.1%)	1 (6.7%)	34 (55.7%)	
First RT-PCR	test	-positive	10 (14.7%)	10 (90.9%)	0 (0%)	<0.01
		-negative	58 (85.3%)	1 (9.1%)	57 (100%)	
LUS & first RT-PCR		-positive	36 (52.9%)	11 (100%)	25 (43.9%)	<0.01
		-negative	32 (47.1%)	0 (0%)	32 (56.1%)	
LUS Score			N = 76	N = 15	N = 61	*p*
Score 0			1 (1.3%)	0 (0%)	1 (1.6%)	1.00
Score 1			34 (44.7%)	1 (6.7%)	33 (54.1%)	<0.01
Score 2			17 (22.4%)	6 (40.0%)	11 (18.0%)	0.1
Score 3			24 (31.6%)	8 (53.3%)	16 (26.2%)	<0.05
Pathological LUS sign						
B Lines ≥3			61 (80.3%)	15 (100%)	46 (75.4%)	<0.05 *
White Lung			32 (42.1%)	10 (66.7%)	22 (36.1%)	<0.05
Thickened Pleura			75 (98.7%)	15 (100%)	60 (98.4%)	1
Fragmented Pleura			40 (52.6%)	14 (93.3%)	26 (42.6%)	<0.01
Consolidation			24 (31.6%)	8 (53.3%)	16 (26.2%)	<0.05
Pleural Effusion			15 (19.7%)	1 (6.7%)	14 (23.0%)	0.17

A *p*-value < 0.05 was defined as significant, * indicate Fischer’s exact test with zero counts, see limitation.

## Data Availability

The LUS datasets (recorded frames) of Figure 2 are available online (see Appendix A) and for further request from the corresponding author.

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
