# Peer review of "Significance of Lung Ultrasound in Patients with Suspected COVID-19 Infection at Hospital Admission"

_diagnostics, 2021, doi:10.3390/diagnostics11060921_

Round 1
Reviewer 1 Report
This study evaluated 76 patients suspected for COVID-19 with lung ultrasound within 48 hours from admission. 34/35 patients who were negative at LUS, were also COVID-19 negative, so Authors conclude that n absence of LUS-specific findings, COVID19 disease can be likely excluded. The topic is of great interest, although there is already a large number of similar papers.
There are some points to be raised:
- LUS may have different accuracy according to the clinical picture of the patients. The clinical characterization here is not so detailed. In particular, no info about oxygen saturation is provided. Were patients only subjectively dyspnoic or were they in respiratory failure? The sentence <In absence of LUS-specific findings COVID19 disease can be likely excluded> is quite strong and, again, depends on patients’ symptoms. This should be better clarified.
- To understand the real added valut of LUS, Authors should explain in how many patients a correct diagnosis could have been done by adding LUS, compared to the standard evaluation.
- “B-line propagation” is not quite used as a term in LUS. Please, just use “B-lines” or explain better what you mean.
- You state that one investigator experienced in ultrasound diagnostics did the exam. What do you mean by “experienced”? Then, what was the specialty of this colleague?
- The overall setting is not so clear: patients were scanned in a hospital ward, but were they referred from the emergency department? What was the mean (or probably bettter, the median and IQR) time lag between admission and LUS in your population? Were more patients scanned close to the 48th hour or to the first hours of admission?
- The LUS protocol you cite in the reference is in German. I suggest an English version.
- Please, specify whether you put the probe and scanned only the specific points of your scanning scheme or fi you scanned the whole thoracic areas.
- Please, specify which ultrasound setting did you use with you linear probe.
- The choice of a linear probe, which is less common than the convex probe, should be explained.
- The term “Pleural Thickening” is not correct. The pleura appears as “thickened” on US, but it is not really thickened.
- The two main papers addressing a very similar situation are not cited not discussed (one is quite recent, but one is less recent). I suggest comparing your results with Pivetta E et al Ann Emerg Med. 2021 Apr;77(4):385-394 and Volpicelli et al Intensive Care Med. 2021 Apr;47(4):444-454.
Author Response
Dear Reviewers,
Thank you very much for the evaluation of our manuscript. Please find enclosed a revised version which has been changed as recommended with a point-by-point account of how we have addressed the issues raised by the reviewers. In addition the grammar has been corrected by an English native colleague. We appreciate the constructive comments which have improved the manuscript and hope that the revisions have enhanced our manuscript sufficiently to be acceptable for publication.
Best Regards
CA
Reviewer 1
- LUS may have different accuracy according to the clinical picture of the patients. The clinical characterization here is not so detailed. In particular, no info about oxygen saturation is provided. Were patients only subjectively dyspnoeic or were they in respiratory failure?
Response: We partly agree.
The inclusion criteria of the study is a general suspicion for a COVID19 infection based on the assumptions about COVID19 at the pandemics beginning. Therefore it was very widely defined (section 2.1), any unclear infect focus, fever >38, cough dyspnoea of any cause ect. Patients with acute severe respiratory failure were not present, also patients requiring immediate ICU care, such as with acute ARDS were excluded due to immediate ICU transfer (see section2.1 page 2 line 34-35).
The definition of dyspnoea is a subjective anamnestic description of the patient, which can be caused by several underlying pathological conditions, such as COPD (1/3 of patients (Tab1)), adiposity, or due cardiopulmonary decompensation and pneumonia related hypercapnia or decreased lung compliance. An oxygen saturation not primary estimates the degree of dyspnoea. Furthermore several studies investigation COVID patients used the same definition (Volpicelli et al 2021 Intensive Care Med page 3 line 2-3).
However, the blood gases were measured for all patients at admission. This revealed that there is no statistical significance between pO2 (77mmHg vs 70mmHg) as well as CO2 (33,3 vs 32,3 mmHg) for COVID neg vs positive group. Therefore the parameters were not presented. However, we realised thanks to the critics, a valuable information that symptomatic COVID screening including BGA is not suitable. Therefore the information about the not existing significance of BGA is added (see page 4 line 11).
- The sentence <In absence of LUS-specific findings COVID19 disease can be likely excluded> is quite strong and, again, depends on patients’ symptoms. This should be better clarified.
Response : We agree
The key finding of our study was the high NPV, which was also one of the highest reported for LUS. If cases are suspicious for COVID infection and negative LUS findings are present, a high likelihood exists that the infection- disease is not present. The strength of LUS as a POC diagnostic is the ability to exclude (or rule out) the disease and not to diagnose it. In addition reviewer2 found this findings particular very valuable. We are aware of the reviewers concerns and have revised the statement and reduced its strength. “The high negative predictive value of LUS is helpful to rule out the disease.”
See Conclusion as well as abstract (page1 line 22, page 9 line 20).
- To understand the real added valut of LUS, Authors should explain in how many patients a correct diagnosis could have been done by adding LUS, compared to the standard evaluation.
Response: We partly agree.
The study represents the clinical conditions at the pandemic beginning, were PCR test are not reliable performed and delayed for days. As our data show the standard evaluation, based on symptomatic at anamnesis is insufficient to rule out the disease, therefore overloading COVID suspect wards. This was the situation we were faced (see introduction). The value of LUS regarding how many patients had correct diagnosis is addressed in the abstract “In 34/35 patients who were assessed as LUS negative, no COVID19 disease was detected during the hospitalisation.” as well as results (see section 3.2 Table 3) and discussion (page 7 line 7-11). As mentioned in question1, LUS is not suitable for diagnosis of SARS COV2 infection, but helpful in ruling out the disease. Therefore the question is confusing, since a diagnosis of the disease would let to a correct classification of 14 out of 15 COVID positive cases by LUS, whilst 27 out of 61 COVID negative cases would diagnosed as well as COVID positive by LUS.
- “B-line propagation” is not quite used as a term in LUS. Please, just use “B-lines” or explain better what you mean.
Response: We agree
See page 2 l12; page 10 line 4
- You state that one investigator experienced in ultrasound diagnostics did the exam. What do you mean by “experienced”? Then, what was the specialty of this colleague?
Response: We agree
Thank you for this hint. The examiners experience of six years in thoracic and vascular sonography as well as the type of theoretical and practical training prior to the study is more detailed described. See section 2.2 page 2 line 38-41.
- The overall setting is not so clear: patients were scanned in a hospital ward, but were they referred from the emergency department?
Response: We agree.
The patients were scanned in the COVID suspect ward in an isolation room. This was not so clear for the reader, thank you for this hint. The aspect is better described see section 2.2 page 2 line42-43; section 3.2 page 5 line 4-5.
5.1 What was the mean (or probably bettter, the median and IQR) time lag between admission and LUS in your population?
Response: We agree.
The median time between admission and LUS was 18h 05’ [9h 48’ 23h 30’] with no statistical significance between COVID – non diseased group (p=0.11). This was not so clear for the reader, and is a valuable hint. The aspect is added see section 3.2 page 5 line 5.
5.2 Were more patients scanned close to the 48th hour or to the first hours of admission?
We agree.
As seen in the previous answer most patients (75%) were scanned by LUS in the first day of admission. The aspect is more detailed outlined see discussion page 9 line 7-8.
- The LUS protocol you cite in the reference is in German. I suggest an English version.
We partly agree:
At that early time of pandemic no international standard has been published. The DEGUM protocol is a wide reference of the German speaking sonographic societies (incl swiss and austria) and in the early phase of the pandemic available, one study author is also co-author of the proposed COVID DEGUM protocol. We are aware of the problematic issue for the reader and added the international reference of Soldati et al 2020 [16] which is very similar except for the scan of 3 instead of two vertebral areas. See section 2.2 page 2 line 45
- Please, specify whether you put the probe and scanned only the specific points of your scanning scheme or fi you scanned the whole thoracic areas.
Response: We agree
The probe was placed in a representative intercostal space, between the ribs, in the specific areal recommended by the reference. Within of those, the probe was slightly moved to get an overview of the areal, and suspicious or pathological artefacts documented.
There was no continuous scanning of the entire thoracic area, which is impractical. The methodology of the scanning process have been more detailed outlined, see section 2.2 page 2 line 45-47.
- Please, specify which ultrasound setting did you use with you linear probe.
Response: We agree:
The scanner specific pre-set for Lung Ultrasound was used. Herein the operator adjusted a single focal depth to the pleural line and a depth of 5-10cm from the pleural depth with a high frame rate with an median MI of 1,1 [0,93 – 1,3]. No THI, compounding was used. These settings were according to the referenced recommendations and general LUS guidelines. These settings and procedure were more detailed described, see section 2.2 page 3 line 5-10.
- The choice of a linear probe, which is less common than the convex probe, should be explained.
Response: we partly agree.
The linear probe are recommended as well as curved probes without further preferences (15 16) some older LUS recommendations even prefer phased arrays which are not so appropriate. It seems that there is no consensus about the preferred probe for LUS and in particular COVID LUS. It has been intensively discussed in our DEGUM LUS meetings without clear conclusion. However, slight preference for linear probes was always present. Limitations of the linear probe arises in obese patients where the lower frequencies of the curved, abdominal probe provide ab deeper penetration (see section 2.2 page 3 line 4).
From our experience the linear probes is the first choice, due to superior imaging of pleural irregularities, which are the most significant LUS artefact of COVID pneumonia. Our preferences are micro curved high frequency (5-10MHz) probe, however this was not available on the used scanner on COVID station.
Our findings and motivation for the linear probe are more detailed discussed see page 8 line 48-49.
- The term “Pleural Thickening” is not correct. The pleura appears as “thickened” on US, but it is not really thickened.
Response: We agree
The term “Pleural Thickening” is inadequate as describes a sonomorphological feature, not an anatomic thickening. However, this phrase is adapted from CT imaging and is used in several well established LUS references in the same way. (e.g. Salehi et al 2020 AJR; Lichter et al 2020 Intens Care Med) . The term has been changed see Fig 2 , page 8 line 5 line 45, page 6 line 4.
Thank you for this hint, which we will further keep in mind when working on general standardisation of LUS terminology, which is recently an ongoing process.
- The two main papers addressing a very similar situation are not cited not discussed (one is quite recent, but one is less recent). I suggest comparing your results with Pivetta E et al Ann Emerg Med. 2021 Apr;77(4):385-394 and Volpicelli et al Intensive Care Med. 2021 Apr;47(4):444-454.
Response: We agree.
Thank you for bringing up this new studies related to the subject. At the time of writing the manuscript these literature were not accessible (Feb-March 21). However the content is relevant and should be added. In addition a new related work important for discussion from Narinx et al [20] is added.
One of the major differences of the new references are different inclusion criteria, as well as LUS artefact definition and the blinded status of the observer. In our study the LUS analysis was performed blinded in absence of knowledge of patients RT-PCR status as well as anamnesis, which can cause a significant bias.
Furthermore a case wise combination of first RT-PCR and LUS haven’t been have been performed in any relevant literature so far.
The literature is added to the discussion section and compared regarding their similar findings of sensitivity as well as differences due to other inclusion criteria as well as pathological LUS definitions. See discussion page 8 line 1-11.
In contrast to the new references, our study performed an analysis of LUS features depending on disease presents. Therefore presenting additional information, necessary to define more COVID specific scoring in the future. In summary there are several literature reflecting the aspect of early LUS at admission for diagnosis of COVID19. The Study inclusion criteria, observer, population condition, as well as time point in the pandemic vary over for each of those. Therefore we believe our (small) study reflects a valuable contribution given the very specific situation at the pandemic beginning. This is essential for future analysis together with upcoming studies to find optimal settings and LUS artefacts definitions to make LUS a frontline POCUS imaging modality to manage future pandemic situations.
Reviewer 2 Report
The topic of this study is obviously interesting. The paper is well done and very clear.The tables, figures, and supplementary material are appropriate. The references are exhaustive.
The most important finding that emerges from this study and that the authors stress in a very pertinent way is the high negative predictive value of lung ultrasound. My field experience in the pandemic period highlighted the importance in clinical practice of the very high negative predictive value of lung ultrasound. This finding was little or not at all stressed by the numerous studies published during this period on the role of LUS. The awareness of the high negative predictive value of LUS is certainly the datum that, moreover, makes possible more adequate and appropriate organizational models, as rightly underlined by the authors.
Author Response
Dear Reviewer
Thank you very much for the evaluation of our manuscript. Please find enclosed a revised version which has been changed as recommended with a point-by-point account of how we have addressed the issues raised by the reviewers. In addition the grammar has been corrected by an English native colleague. We appreciate the constructive comments which have improved the manuscript and hope that the revisions have enhanced our manuscript sufficiently to be acceptable for publication.
Best Regards
CA
Reviewer2:
The most important finding that emerges from this study and that the authors stress in a very pertinent way is the high negative predictive value of lung ultrasound. My field experience in the pandemic period highlighted the importance in clinical practice of the very high negative predictive value of lung ultrasound. This finding was little or not at all stressed by the numerous studies published during this period on the role of LUS. The awareness of the high negative predictive value of LUS is certainly the datum that, moreover, makes possible more adequate and appropriate organizational models, as rightly underlined by the authors.
Response: We agree.
Thank you for valuing the key findings of our study. The aspect of the high NPV of LUS and its clinical relevance are more detailed addressed due to reviever1 comments. See conclusion and page 8 line 15-17
Round 2
Reviewer 1 Report
The Authors sufficiently addressed most of the issues raised.
However, the term "thickened pleura" is still present (Discussion).
I also would change the term "LUS artefacts", since only B-lines are more clearly artefacts; I'd rather use "LUS signs".
Author Response
Reviewer 1
However, the term "thickened pleura" is still present (Discussion).
Response: We partly agree.
As correctly addressed in the first review round by reviewer 1, the term has been changed from “pleural thickening” to thickened pleura / pleural line.
“The term “Pleural Thickening” is not correct. The pleura appears as “thickened” on US, but it is not really thickened.”
In order to avoid confusions for the reader, we defined the term “thickened pleura” at the first time in the scoring definition, see page 3 line 15-16. Because it is important to define it as an LUS sign.
Score 1: : B-lines (number >3) or white lung syndrome (WLS), irregular pleural line or when pleura appears as thickened on sonography (thickened pleural)
Furthermore the term is formulated in detail whenever possible. See page 8 line 46. (Sonographic appearance of thickened pleura, …. )
I also would change the term "LUS artefacts", since only B-lines are more clearly artefacts; I'd rather use "LUS signs".
Response: We agree.
Thank you for this advice, artefacts are more common associated with B Lines or WLS. LUS characteristics including B Lines are typical found to be described as feature or sign.
The manuscript was changed in a way that artefacts are replaced with signs or feature when the expression is not B-lines related. See page 2 line 43, table 3, page 7 line 40, page 8 line 27, 38